# Clinical and Genetic Heterogeneity of Factor XI Deficiency: Insights from a Southern Italian Cohort

**DOI:** 10.3390/ijms26188807

**Published:** 2025-09-10

**Authors:** Rosa Santacroce, Giovanna D’Andrea, Giovanni Luca Tiscia, Giuseppe Lassandro, Maria d’Apolito, Doris Barcellona, Patrizia De Bonis, Francesco Marongiu, Paola Giordano, Elvira Grandone, Maurizio Margaglione

**Affiliations:** 1Medical Genetics, Department of Clinical and Experimental Medicine, University of Foggia, 71122 Foggia, Italy; rosa.santacroce@unifg.it (R.S.); giovanna.dandrea@unifg.it (G.D.); maria.dapolito@unifg.it (M.d.); 2Thrombosis and Haemostasis Unit, Fondazione IRCCS “Casa Sollievo della Sofferenza”, Viale Cappuccini, 71013 San Giovanni Rotondo, Italy; g.tiscia@operapadrepio.it (G.L.T.); p.debonis@operapadrepio.it (P.D.B.); e.grandone@operapadrepio.it (E.G.); 3Department of Biomedical Sciences and Human Oncology, Pediatric Section, University of Bari “A. Moro”, 70126 Bari, Italy; giuseppelassandro@live.com (G.L.); paola.giordano@uniba.it (P.G.); 4Department of Medical Science and Public Health, University of Cagliari, 09124 Cagliari, Italy; doris.barcellona@unica.it (D.B.); marongiu@medicina.unica.it (F.M.); 5Departmental Unit of Thrombosis and Haemostasis, Azienda Ospedaliero-Universitaria di Cagliari, 09124 Cagliari, Italy; 6Obstetrics and Gynecology Department, University of Foggia, 71122 Foggia, Italy; 7Department of Obstetrics, Gynaecology and Perinatal Medicine, First I.M. Sechenov Moscow State Medical University, 119991 Moscow, Russia

**Keywords:** FXI deficiency, gene variant, levels, bleeding, phenotype

## Abstract

Factor XI (FXI) deficiency, or hemophilia C, is a rare bleeding disorder resulting from reduced levels or dysfunctional FXI protein due to mutations in the *F11* gene. This study investigated the correlation between FXI activity levels, *F11* genotype, and bleeding phenotypes. Clinical and genetic characteristics of 93 individuals from southern Italy diagnosed with congenital FXI deficiency, including 39 index cases and their relatives, were evaluated. FXI:C plasma levels were measured. Sanger sequencing of *F11* was performed, and the pathogenicity of variants identified was assessed using in silico tools. FXI activity levels ranged widely (1–69%), with most cases being heterozygous and showing moderate deficiency. Only 12 individuals had severe FXI deficiency, typically associated with homozygosity or compound heterozygosity. Bleeding symptoms varied from mild to severe and occurred in 31% of subjects, though only a minority of those with severe deficiency experienced spontaneous or surgery-related bleeding. Sanger sequencing revealed 24 distinct *F11* gene variants, predominantly missense mutations, with three novel variants (p.Val89*, p.Leu306Pro, and p.Trp515Gly). Common mutations included p.Glu135* and p.Glu315Lys. Variants were distributed across the gene, with no domain-specific clustering. No clear genotype–phenotype correlation was observed. FXI levels alone did not reliably predict bleeding risk, highlighting the influence of additional factors such as age, gender, and clinical history. This study reinforces the allelic and clinical heterogeneity of FXI deficiency and the limited utility of FXI:C levels alone for predicting bleeding severity. Further research is needed to clarify the complex genotype–phenotype relationships in FXI deficiency.

## 1. Introduction

Factor XI (FXI) deficiency, also known as hemophilia C, is a rare bleeding disorder that affects individuals due to reduced levels or impaired function of FXI, a coagulation factor that plays a crucial role in the intrinsic pathway [1]. This deficiency is considered rare, affecting a small proportion of the population. FXI deficiency is most commonly found in individuals of Ashkenazi Jewish descent, with a prevalence of 1 in 1000 individuals [2]. The pathogenesis of FXI deficiency lies in genetic mutations affecting the *F11* gene, which is located on chromosome 4 and encodes the FXI protein. *F11* variants can lead to a large array of alterations in protein synthesis or functionality, resulting in reduced or dysfunctional FXI [3]. Any disruption in the molecular processes involved in the synthesis, post-translational modification, or function of FXI can contribute to deficiency [4]. This may involve abnormalities in protein folding, transportation, or interaction with other clotting factors. The inheritance pattern of severe FXI deficiency is autosomal recessive. This means that an individual has to carry two alleles with a pathogenic variant, usually inherited one from each parent, to manifest the severe form of the disorder. Heterozygotes, individuals with a normal and a mutated copy of the gene, typically do not show symptoms or manifest an injury-related bleeding phenotype but can transmit the variant allele to their offspring.

FXI deficiency exhibits considerable genetic heterogeneity, and a high number of variants have been identified in the *F11* gene. These variations may include missense mutations, nonsense mutations, deletions, or insertions, each contributing to the varied clinical manifestations observed in affected individuals. The type and location of these gene variants can influence the severity of the deficiency and the associated bleeding tendencies.

Bleeding in FXI-deficient patients can range from mild to severe, with symptoms including prolonged bleeding after injury or surgery, nosebleeds, and easy bruising [5]. However, the severity of bleeding can vary greatly among individuals with FXI deficiency, making it difficult to predict the risk of bleeding in each patient [6].

Understanding the predictability of bleeding in patients with FXI deficiency is crucial for effective patient management and the development of appropriate treatment strategies [7].

FXI protein structure is a homodimer, with each subunit containing four apple domains (A1 to A4) and a trypsin-like catalytic domain. *F11* mutations have been identified in all domains, suggesting that distinct structure–function relationships of different domains of the FXI could play a role in leading to the clinical phenotype [1].

In this study, we have evaluated the incidence and characteristics of gene variants affecting *F11* in a cohort of patients with FXI deficiency recruited in reference hospitals in southern Italy.

## 2. Results

### 2.1. Clinical Features of Cases with FXI Deficiency

In the present study, a cohort of 93 individuals (37 men and 56 women) with a diagnosis of FXI deficiency from 39 unrelated families was recruited in four southern Italian hospitals over a 15-year period (2007–2022). Mean age at enrollment was 31.6 years (range: 1–82). The large majority of individuals (81/93: 87%) showed a moderate FXI deficiency, suggestive of heterozygous defects, but 12 cases had severe FXI deficiency compatible with homozygosity or compound heterozygosity (Figure 1). Mean FXI activity levels were 35.1 IU/dL (range: 1–69).

Most of the 39 index cases were referred for abnormal values during preoperative screening and check-ups (n = 21, 54%). The remaining index cases were referred for bleeding symptoms. Among the cases where abnormal bleeding occurred (n = 18, 46%), eight resulted from bleeding after surgery or trauma. In addition, four women presenting with a personal history of menorrhagia were diagnosed with a FXI deficiency, whereas four cases suffered from recurrent nosebleeds and two from spontaneous ecchymoses. Among the nine index cases presenting with severe FXI deficiency, three bled: an 8-year-old female had bleeding after surgery; a 24-year-old woman showed a history of menorrhagia; and a 49-year-old woman who had repeated hemorrhagic episodes after surgery or operative dentistry procedures.

Fifty-four relatives with reduced FXI levels were identified. All relatives but four showed a moderate FXI deficiency. Among them, 9 out of 50 with a moderate deficiency (18%) and 2 out of 4 with a severe deficiency (50%) suffered from bleeding after surgery or trauma (n = 8), recurrent nosebleed (n = 2), or menorrhagia (n = 1).

### 2.2. Molecular Characterization

Sanger sequencing of *F11* identified 24 potential causative variants. All were single-nucleotide variations (SNVs), mainly causing missense changes (n = 17). In addition, three nonsense variants, three variants affecting splicing, and a small deletion were detected. As expected, variations dispersed over the entire *F11* gene with no evidence of clustering at specific coding and non-coding regions (Table 1) but primarily involved residues in the Apple 2, Apple 4, and serine protease domains (Figure 1). Bleeding tendency was equally distributed among subjects carrying variants in different FXI protein domains, and no significant genotype–phenotype correlation was detected.

As expected, most of the cases were heterozygotes (79/93: 88%), whereas homozygous and compound heterozygous were four (4/93: 1%) and ten (10/93: 10%), respectively (Table 1). Heterozygotes presented approximately four-fold higher mean plasma FXI:C levels than those recorded in homozygotes or compound heterozygotes, being 39 IU/dL (range: 18–79) and 10.6 IU/dL (range: 1–36), respectively. Among the 25 FXI defective cases presenting with a hemorrhagic phenotype, two were compound heterozygotes, and all others were heterozygotes.

Of the 24 different FXI gene variants identified, 21, 20, and 18 have been previously reported in subjects with a FXI deficiency and were included by 1 June 2025 in the EAHAD coagulation Factor XI variant, in the UCL Factor XI Gene (*F11*), and in the HGMD databases, respectively.

In the gnomAD v4.1.0 database, 15 of these variants had a germline classification and were reported in the ClinVar archive as pathogenic (n = 7), likely pathogenic (n = 3), with conflicting interpretations (n = 3), or of uncertain significance (n = 2).

Among gene variants of uncertain significance, the p.Asp34His substitution has been suggested to interfere with chain folding [11]. The p.Arg162Cys substitution replaces a polar residue Arginine containing an electrically charged side chain with a sulfur-containing neutral and slightly polar amino acid, Cysteine. Among gene variants with conflicting interpretations, the substitution of a Threonine with a Methionine at residue #141 results in the introduction of a hydrophobic sulfur atom and the loss of a hydrophilic hydroxyl group [12]. The p.Thr150Met substitution replaces a hydrophilic residue with a hydrophobic one. Finally, the p.Glu565Lys substitution has been suggested to affect *F11* mRNA splicing and induce the exon 13 skipping [13,14].

In the gnomAD v4.1.0 database, the p.Cys136Arg and the p.Arg497Gln variants were present at a very low frequency (1.86 × 10^−6^ and 9.30 × 10^−6^, respectively). The first missense variation results in the substitution of a positively charged amino acid for a neutral one containing a sulfur atom. The latter causes the substitution of an Arginine for a Glutamine, which has no ionizable side chain.

Two variants with unknown implications on gene transcription were located in the exon–intron boundaries. The first was the c.595+3A>G transition, which occurred in intron 6. In silico investigation of the effect of the substitution predicted a loss of efficiency of the splicing machinery (score of the donor site changing from 0.98 to 0.56), causing the skipping of the affected exon 7. The second was the c.1717-2A>G transition and occurred in the intron 14. In silico investigation of the effect of the substitution predicted a deleterious effect with the suppression of the acceptor site (score changing from 0.98 to 0.0), causing the skipping of the affected exon 14.

The c.G325A (p.Ala109Thr) variant has been reported to interfere with the physiological donor splice site, resulting in the skipping of exon 4 [15].

Among the remaining F11 gene variants identified, the p.Ala561Asp was identified together with the p.Glu315Lys in a woman and her daughter. In both cases, FXI plasma levels were comparable (33.6 IU/dL and 36.3 IU/dL). Inheritance of both variants suggested that they are in linkage, and the moderate reduction in FXI plasma levels indicated the lack of an important effect of the p.Ala561Asp variant on the protein functionality. The p.Trp515Gly replaces a large aromatic and hydrophobic residue with a small, non-polar, and hydrophilic amino acid. The substitution was predicted to produce a pathogenic effect (suggested classification: Varsome: PP3; Franklin: Likely pathogenic).

The remaining three FXI gene variants (p.Val89*, p.Leu306Pro, and p.Trp515Gly) were new and previously unreported. The p:Val89* variant causes the appearance of a stop codon and is predicted to severely affect protein expression. In keeping with this, the index case showed reduced plasma FXI levels (42.3%). Analysis of the p.Leu306Pro variant using computational prediction tools (Table 2) showed an extremely low frequency in gnomAD population databases (PM2: Pathogenic Moderate) and supported a deleterious effect on the gene (PP3: Pathogenic Moderate). In addition, the missense variant occurred in a gene with a low rate of benign missense mutations and for which a missense mutation is a common mechanism of a disease (PP2: Pathogenic Supporting). The p.Trp515Gly replaces a large aromatic and hydrophobic residue with a small, non-polar, and hydrophilic amino acid. The substitution was predicted to produce a pathogenic effect (suggested classification: Varsome: Pathogenic Supporting; Franklin: Likely pathogenic). The in silico analysis using the MISSENSE3D tool (https://missense3d.bc.ic.ac.uk/ as assessed on accessed on 1 June 2025) detected structural damage in both missense variants. p.Leu306Pro substitution triggers a disallowed phi/psi alert. The phi/psi angles are in the favored region for the wild-type residue but the outlier region for the mutant residue. The p.Trp515Gly substitution disrupts all H-bonds formed by a buried TRP residue (RSA 6.6%). In addition, it leads to the expansion of cavity volume by 113.832 Å^3^.

For each genetic variant, the following information is shown: variant allele frequency according to gnomAD exome/genome database, DyneMut2_ (ΔΔGStability), Predicted Stability Change, effect on Protein Structure, ACMG annotation, ACMG Supporting Criteria. VUS: variant of uncertain significance.

To analyze the movement and flexibility of the mutant p.Leu306Pro and p.Trp515Gly proteins, we used the AlfaFold prediction structure (AF-P03951-F1) of the human Coagulation factor XI as a template and DynaMut2, a web server that combines Normal Mode Analysis (NMA) methods. DynaMut2 predicted a significant effect of both p.Leu306Pro and p.trp515Gly missense variations on protein stability with a Predicted Stability Change (ΔΔGStability) of −0.32 kcal/mol and −2.94 kcal/mol, respectively, indicating a destabilizing effect. In this tool, ΔΔG ≥ 0 is considered stabilizing, and ΔΔG < 0 is considered destabilizing (Table 2). The modeling identifies both model modifications in a mutant structure, allowing the formation of new extra bonds between the mutated protein and nearby residues or the loss of wild-type bonds (Figure 2).

### 2.3. FXI Deficiency and Bleeding

To investigate the presence of a genotype–phenotype correlation, the incidence of a higher bleeding rate was evaluated in homozygous or composite heterozygous patients compared to patients with a heterozygous genotype. Of the 14 patients with a homozygous or compound heterozygous genotype, 5 experienced a hemorrhage (35.7%). Twenty-four of the seventy-eight heterozygous patients experienced a hemorrhage (30.8%; p: n.s. Fisher Exact test).

In index cases, the most common *F11* gene variants identified were p.Glu135* (also known as Jewish mutation type II; 23%, 9/39), p.Glu315Lys (21%, 8/39), and p.Phe301Leu (also known as Jewish mutation type III; 10%, 4/39). Analysis of genotype–phenotype co-segregation of heterozygous carriers of one of the three most frequent variants of the *F11* gene showed that only about half of the patients had experienced at least one bleeding episode. Indeed, five out of the 15 heterozygotes for the Glu135* allelic variant, five of the 14 heterozygotes for the Glu315Lys variant, and none of the four carriers of the Phe301Leu variant have ever had a bleeding episode.

Furthermore, the mean bleeding score assessed by ISTH-BAT [10] was not increased in cases with defective FXI (mean 1, range 0–6) compared to historical healthy control populations [16,17]. No linear relationship was observed between bleeding scores and plasma FXI values (Pearson correlation coefficient: −0.18; p: n.s.). Finally, the mean ISTH-BAT value was not different between heterozygous carriers and homozygous or compound heterozygous patients with deficient FXI (1.64 ± 2.37 vs. 0.87 ± 1.49; p: n.s. Student’s *t*-test).

## 3. Discussion

Circulating FXI levels play a significant role in determining the bleeding tendencies of individuals with FXI deficiency. Several studies have demonstrated a correlation between lower FXI levels and an increased risk of bleeding events. Lower levels of FXI are associated with an increased risk of bleeding, while higher levels may provide some degree of protection against bleeding episodes. Patients with FXI levels below 30 IU/dL have been more likely to experience spontaneous bleeding [18]. In addition, FXI levels have been reported to serve as a reliable predictor of bleeding severity in FXI-deficient patients [19]. We report a higher bleeding prevalence among subjects with a severe FXI deficiency. These findings highlight the importance of monitoring FXI levels to better understand bleeding predictability in this population. However, it is important to note that FXI levels alone do not fully predict the severity of bleeding in FXI-deficient patients. In the current context, even in severe cases of FXI deficiency, the bleeding tendency is generally mild or moderate, while spontaneous bleeding is infrequent. We confirm a high variability in bleeding tendency, which was barely associated with FXI activity. In fact, 29 out of 93 individuals showed bleeding episodes, and only 5 out of these had shown a severe deficiency.

Potentially pathogenic variants were identified in all subjects presenting with FXI deficiency. A total of 24 different *F11* gene variations were identified, and all were single-nucleotide variants. Most of them were previously reported, while three were described for the first time (p.Val89*, p.Leu306Pro, and p.Trp515Gly). Pathogenic variants were equally distributed throughout the entire *F11* gene. In keeping with other studies in Italian patients [20,21], some *F11* gene variants were prevalent, i.e., p.Glu135* (also known as Jewish mutation type II). All these results demonstrate the high allelic heterogeneity of FXI deficiency.

The novel p.Ala561Asp variant was found both in a young woman and in her daughter in association with the p.Glu315Lys variation. This finding suggests that the two variants form a haplotype and segregate in cis with each other. The p.Glu315Lys variant is reported in the ClinVar archive as pathogenic. In addition, in the present investigation, heterozygotes showed FXI levels similar to those observed in previous studies. All these findings suggest that the p.Ala561Asp variant in cis does not apparently influence FXI plasma values or the clinical phenotype. However, it occurs in a highly conserved sequence—Cysteine–Alanine–Glycine—among serine proteases. Missense alterations involving Cysteine–Alanine–Glycine residues of the *F11* gene or other serine proteases of the coagulation system have been described [22].

The most common *F11* gene variants identified were p.Glu135* and p.Glu315Lys, recorded in 21 (14 heterozygotes, 4 compound heterozygotes, and t3hree homozygotes) and 17 individuals (15 heterozygotes, including those also carrying the p.Ala561Asp variant, 1 compound heterozygote, and 1 homozygote), respectively. Overall, a high variability in the expression of the clinical phenotype was observed among heterozygotes with one of these two variants. In fact, 19 out of 29 individuals were asymptomatic, and only 4 of the 19 women suffered from menorrhagia. There are currently limited data on bleeding complications in women with FXI deficiency. In our cohort of women (25 probands and 28 relatives), we recorded five menorrhagia (9.4%). This prevalence did not differ from that recorded in apparently normal women, in which up to one fourth may experience some form of heavy menstrual bleeding [23]. Overall, all these data indicated that bleeding tendency is equally distributed among subjects carrying variants in different domains, and no significant genotype–phenotype correlation was detected.

FXI belongs to the contact phase of the coagulation pathway, and its activation can be directly mediated by kallikrein [24]. Therefore, it can be argued that alterations in the contact phase may modulate the risk of bleeding in patients with FXI deficiency. Furthermore, it has been suggested that surgery-related bleeding complications occur mainly at sites with increased fibrinolytic activity [4,25]. This may explain why the clinical symptoms of FXI deficiency are typically related to lesions or specific surgical procedures, such as circumcision and urogenital surgery, or in females during the reproductive period [26,27,28].

Risk stratification models have been developed to enhance the accuracy of bleeding risk assessment in patients with rare bleeding disorders. The ISTH bleeding assessment tool and the EN-RBD bleeding score are examples of risk prediction models that have been widely used in clinical practice to guide management decisions for patients with rare bleeding disorders [7,17,29,30]. These models take into account various clinical parameters, including plasma levels of some coagulation factors, to provide a more comprehensive evaluation of bleeding tendencies. Other factors, such as age, gender, and the presence of comorbidities, can also influence the bleeding tendencies in these individuals [1]. By integrating FXI levels with other relevant factors, such as age, bleeding history, type of surgery, and comorbidities, these models hopefully may offer a more accurate prediction of bleeding risk.

## 4. Limitations

The present study has some limitations because longitudinal functional data are lacking. Due to the sample size, we did not explore and thus cannot rule out whether the relationship between FXI deficiency and outcomes differs by sex. In addition, drugs and comorbidities might have somewhat interfered with FXI levels, though many potential cofounders were excluded. However, the present study confirms that FXI activity levels poorly correlate with bleeding phenotypes.

## 5. Materials and Methods

### 5.1. Case Index and Relatives

A total of 82 individuals belonging to 28 independent families and an additional 11 unrelated index patients with congenital FXI deficiency were recruited at four reference centers for thrombotic and hemorrhagic disorders. The inclusion criteria encompassed patients with a congenital FXI deficiency, characterized by a factor XI activity of <70 IU/dL. Exclusion criteria included acquired FXI deficiency, liver failure, and consumptive coagulopathy. Severe FXI deficiency was defined as an activity level of <20 IU/dL [25]. Partial FXI deficiency was diagnosed among individuals who had FXI activity levels of 20 to <70 IU/dL. Data were collected from medical records, including clinical, biological, and therapeutic data at the time of diagnosis and subsequent evolution. Clinical and genetic investigations were performed in accordance with the Helsinki declaration and based on written informed consent for clinical and genetic testing. Written informed consent was requested from their legal representatives for subjects under 18 years. All data presented in this manuscript were properly anonymized. The study was approved by the local ethical committee (protocol code 3261/CE/20) on (9 October 2018).

### 5.2. Coagulation Tests

Peripheral venous blood was obtained from all patients using siliconized glass tubes containing sodium citrate anticoagulant. The upper layer of poor-platelet plasma was utilized for routine coagulation screening after centrifugation at 3000× *g* for 10 min. The prothrombin time (PT), activated partial thromboplastin time (APTT), fibrinogen, and FXI:C were detected using SynthASil and FXI-deficient plasma, both from Werfen, on the ACL top automatic analyzer (Werfen, Barcelona, Spain). The detection limits of the FXI:C assay were <1 U/dL. Reference intervals were obtained from the plasma of 100 healthy blood donors from the local healthy population.

### 5.3. Genetic Investigation

According to the manufacturer’s instructions, leukocytes from participants were utilized for extracting genomic DNA via QIAamp DNA Blood Kits (GIAGEN, Hilden, Germany). All 15 exons of the *F11* gene, along with exon/intron boundary regions, were amplified using specific primers, and PCR products were then sequenced using BigDye Terminator v.3.1 (Thermo Fisher Scientific, Waltham, MA, USA), according to standard protocols [31]. PCR products were purified with PCR multiwell 96-well plates (Merck Millipore ldt., Tullagreen, Ireland). The purification protocol involves adding 55 μL H_2_O to the PCR product and transferring it to the plate, leaving the plate at room temperature on the vacuum pump for 8 min. Afterwards, add 26 μL H_2_O and let the plate shake for 10 min at 500rpm. At the end of the process, we obtained the purified product. Then, the sequencing reaction was performed by BigDye Terminator v3.1 Cycle Sequencing Kit and was analyzed on an ABI 3130xl DNA Sequencer (Applied Biosystems, Norwalk, CT, USA). The sequencing reaction mixtures contained 2 μL of BigDie Terminator v1.1, 3.1 5X sequencing buffer; 1.2 μL of the same primer used in PCR previously diluted 1:100; 0.5 μL of BigDie Terminator v3.1 Cycle Sequencing RR-100; and 0.5 μL of purified PCR in a final volume of 10 μL H_2_O. Sequencing files were processed using Sequence Analysis Software v6.0 (Applied Biosystems, Norwalk, CT, USA) and were aligned and analyzed using Sequencer 4.7 Software. Prioritized Variant was validated in probands by Sanger sequencing and then studied in additional affected family members when DNA was available to perform segregation analysis.

### 5.4. In Silico Analysis of Pathogenicity

Population data were obtained from the Genome Aggregation Database (gnomAD; https://gnomad.broadinstitute.org/ accessed on 1 June 2025). ClinVar (https://www.ncbi.nlm.nih.gov/clinvar accessed on 1 June 2025), Franklin by genoox (https://franklin.genoox.com/clinical-db/home accessed on 1 June 2025), EAHAD coagulation Factor XI UCL Factor XI Gene (*F11*) (https://dbs.eahad.org/FXI accessed on 1 June 2025), (https://www.factorxi.org/torifxi_reference.html.php accessed on 1 June 2025), and The Human Gene Mutation Database (https://www.hgmd.cf.ac.uk/ac/ accessed on 1 June 2025) were used as tools to sum up actual knowledge about the variants.

To investigate the putative pathogenic effect of new unreported *F11* gene variants, established available bioinformatics tools were used (MISSENSE3D: https://missense3d.bc.ic.ac.uk/ accessed on 1 June 2025).

To explore the potential deleterious impact of the new missense variations, DynaMut2 (https://biosig.lab.uq.edu.au/dynamut/ accessed on 1 June 2025) was then utilized by using the AlfaFold prediction structure (AF-P03951-F1) of a human FXI. DynaMut2 is a web server that can evaluate the effects of mutations on the vibrational entropy variations induced and modifications in protein dynamics and stability, as well as analyze and visualize protein dynamics by sampling conformations.

### 5.5. Statistical Analysis

Absolute numbers, percentage, and median (range) or mean (±standard deviation) were calculated to describe study groups and sub-populations of interest. The significance of any difference in means was evaluated by parametric tests, whereas the significance of any difference in proportions was tested using Fisher’s exact test or by χ^2^ statistics as appropriate. Statistical analyses were performed using SPSS version 11.0 (SPSS Inc., Chicago, IL, USA).

## 6. Conclusions

The data from the present study confirm the wide heterogeneity of clinical and molecular findings in subjects with FXI deficiency and a weak correlation between FXI plasma levels and clinical outcome in FXI-deficient patients. Further studies are needed to better define the genotype–phenotype relationship in subjects with FXI deficiency.

## Figures and Tables

**Figure 1 ijms-26-08807-f001:**
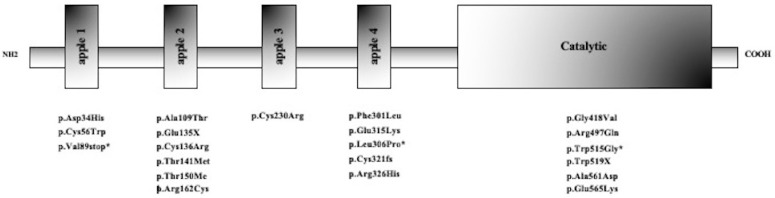
Distribution of single-nucleotide variants along protein domains of FXI. New *F11* gene variants identified are indicated with an asterisk (*).

**Figure 2 ijms-26-08807-f002:**
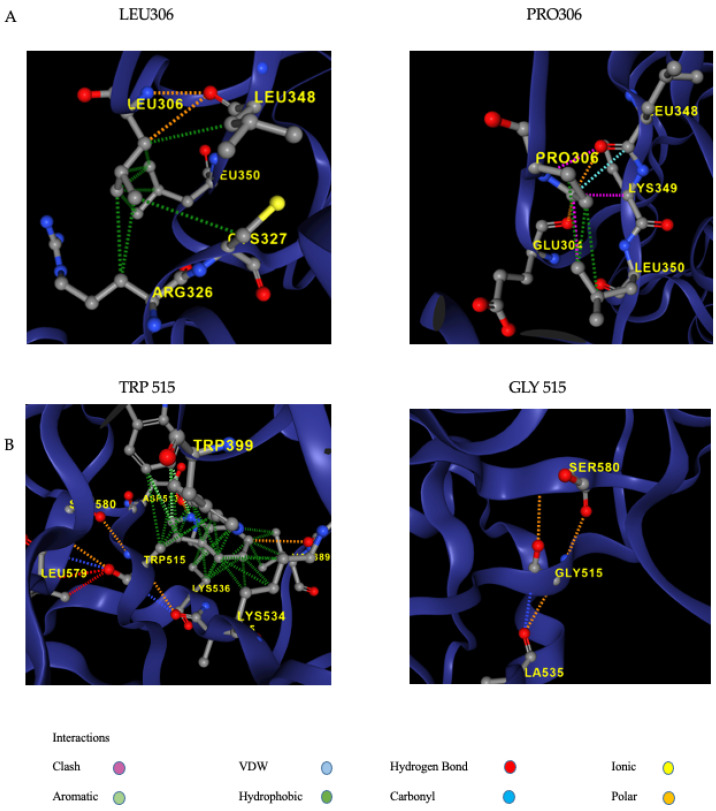
Predicted structures of the wild-type (**A left**: 306L; **B left**: 515W) and variant protein (**A right**: 306P; **B right**: 306G) are compared using the DynaMut2 online server.

**Table 1 ijms-26-08807-t001:** Clinical and molecular findings of the families investigated.

Case Index	Sex	Age at the Presentation	FXI Activity (IU/dL)	Variant 1	Variant 2	Symptoms	ISTH-BAT
1	F	21	51	p.Asp34His		Asymptomatic	0
1-1	F	26	47	p.Asp34His		Asymptomatic	0
1-2	F	49	55	p.Asp34His		Asymptomatic	0
2	M	58	43	p.Cys56Trp		Bleeding after surgery	3
2-1	M	32	38	p.Cys56Trp		Asymptomatic	0
2-2	F	25	38	p.Cys56Trp		Asymptomatic	0
3 *	F	56	42	p.Val89stop		Repeated bleeding after surgery	4
4	M	35	32	C.325+1G>A		Epixastis	2
5 (a)	F	1	1	p.Ala109Thr	C.325+1G>A	Asymptomatic	0
5-1	M	31	48	C.325+1G>A		Asymptomatic	0
5-2	F	4	41	p.Ala109Thr		Asymptomatic	0
5-3	F	29	35	p.Ala109Thr		Asymptomatic	0
6 (a)	M	32	39	p.Glu135X		Epixastis	2
7 (a)	F	60	44	p.Glu135X		Repeated bleeding after surgery, Menorrhagia	6
8 (a)	F	26	38	p.Glu135X		Asymptomatic	0
8-1	F	37	34	p.Glu135X		Asymptomatic	0
9	M	16	28	p.Glu135X		Asymptomatic	0
9-1	F	13	25	p.Glu135X		Asymptomatic	0
9-2	M	43	48	p.Glu135X		bleeding after surgery or trauma	5
9-3	F	46	33	p.Glu135X		Asymptomatic	0
10	F	7	4	p.Glu135X	p.Cys321fs	Asymptomatic	0
10-1	F	34	69	p.Cys321fs		Asymptomatic	0
10-2	F	5	3	p.Glu135X	p.Cys321fs	Asymptomatic	0
10-3	F	3	39	p.Glu135X		Asymptomatic	0
11 (b)	F	47	1	p.Glu135X	p.Glu135X	Repeated bleeding after surgery or trauma	6
11-1	M	76	50	p.Glu135X		Asymptomatic	0
11-2	F	74	76	p.Glu135X		Asymptomatic	0
11-3	M	45	1	p.Glu135X	p.Glu135X	Repeated bleeding after surgery or trauma	5
11-4	F	33	52	p.Glu135X		Asymptomatic	0
11-5	M	41	2	p.Glu135X	p.Glu135X	Repeated bleeding after surgery or trauma	5
12 (a)	F	55	1	p.Glu135X	p.Cys136Arg	Asymptomatic	0
12-1	F	57	2	p.Glu135X	p.Cys136Arg	Asymptomatic	0
13	F	26	40	p.Glu135X		Menorrhagia	2
14	F	41	60	p.Glu135X		Menorrhagia	3
15	M	10	22	p-Thr141Met		Asymptomatic	0
15-1	M	54	35	p-Thr141Met		Asymptomatic	0
16 (a)	M	7	34	p.Thr150Met		Epistaxis	2
16-1	M	38	52	p.Thr150Met		Epistaxis	1
16-2	M	46	41	p.Thr150Met		Asymptomatic	0
16-3	M	21	34	p.Thr150Met		Epistaxis	1
17 (a)	F	6	47	p.Arg162Cys		Asymptomatic	0
17-1	F	6	43	p.Arg162Cys		Asymptomatic	0
17-2	F	36	45	p.Arg162Cys		Asymptomatic	0
18 (a)	M	20	34	p.Cys230Arg		Bleeding after surgery	3
18-1	F	50	34	p.Cys230Arg		Asymptomatic	0
19	M	41	44	p.Phe301Leu		Repeated bleeding after surgery	3
20	F	5	46	p.Phe301Leu		Asymptomatic	0
21 (a)	F	8	6	p.Phe301Leu	p.Trp519X	Bleeding after surgery	4
21-1	F	39	40	p.Phe301Leu		Asymptomatic	0
22 (a)	F	14	4	p.Phe301Leu	c.595+3A>G	Asymptomatic	0
22-1	F	43	85	c.595+3A>G		Asymptomatic	0
22-2	M	49	51	p.Phe301Leu		Asymptomatic	0
23 *	M	15	40	p.Leu306Pro		Asymptomatic	0
23-1	F	11	45	p.Leu306Pro		Asymptomatic	0
23-2	F	49	38	p.Leu306Pro		Asymptomatic	0
24	F	3	34	p.Glu315Lys	p.Ala561Asp	Asymptomatic	0
24-1	F	29	36	p.Glu315Lys	p.Ala561Asp	Asymptomatic	0
25	F	37	28	p.Glu315Lys		Menorrhagia	2
25-1	F	36	33	p.Glu315Lys		Bleeding after surgery	2
25-2	M	66	18	p.Glu315Lys		Bleeding after surgery	3
26 (a)	F	24	7	p.Glu315Lys	p.Trp519X	Menorrhagia	3
26-1	F	22	29	p.Glu315Lys		Asymptomatic	0
26-2	M	60	49	p.Trp519X		Asymptomatic	0
26-3	F	46	39	p.Glu315Lys		Repeated bleeding after trauma	4
27	M	16	34	p.Glu315Lys		Asymptomatic	0
27-1	M	45	27	p.Glu315Lys		Asymptomatic	0
28	M	10	31	p.Glu315Lys		Asymptomatic	0
28-1	F	49	37	p.Glu315Lys		Menorrhagia	2
29	M		4	p.Glu315Lys	p.Glu315Lys	Asymptomatic	0
30	F	15	36	p.Glu315Lys		Asymptomatic	0
30-1	M	38	28	p.Glu315Lys		Asymptomatic	0
31	M	6	41	p.Glu315Lys		Asymptomatic	0
31-1	M	40	40	p.Glu315Lys		Asymptomatic	0
32 (a)	F	18	38	p.Arg326His		Asymptomatic	0
32-2	F	44	62	p.Arg326His		Asymptomatic	0
33 (a)	F	29	43	p.Gly418Val		Spontaneous ecchymoses	3
33-1	F	52	45	p.Gly418Val		Repeated bleeding after trauma	3
34	M	10	21	p.Arg497Gln		Asymptomatic	0
35 (a) *	F	40	14	p.Trp515Gly		Asymptomatic	0
35-1	M	17	30	p.Trp515Gly		Asymptomatic	0
35-2	F	15	44	p.Trp515Gly		Asymptomatic	0
35-3	F	43	27	p.Trp515Gly		Asymptomatic	0
35-4	M	46	36	p.Trp515Gly		Asymptomatic	0
35-6	F	69	36	p.Trp515Gly		Asymptomatic	0
36	F	82	40	P.Glu565Lys		Repeated bleeding after surgery	3
37	F	18	29	P.Glu565Lys		Spontaneous ecchymoses	3
37-1	M	57	50	P.Glu565Lys		Asymptomatic	0
38	M	10	30	P.Glu565Lys		Asymptomatic	0
38-1	M	6	28	P.Glu565Lys		Asymptomatic	0
38-2	M	36	28	P.Glu565Lys		Repeated bleeding after surgery—Epistaxis	5
38-3	F	63	39	P.Glu565Lys		Asymptomatic	0
39	M	21	48	c.1717-2A>G		Epistaxis	2
39-1	F	24	42	c.1717-2A>G		Asymptomatic	0

(a): Already described in [8]; (b): already described in [9]. New *F11* gene variants identified are indicated with an asterisk (*). BAT: ISTH-SSC Bleeding Assessment Tool [10]: https://www.isth.org/page/reference_tools accessed on 25 August 2025).

**Table 2 ijms-26-08807-t002:** Information on the new F11 gene variants identified.

GENE	Protein Variation	Frequencies	ClinVar	DyneMut2_ (ΔΔGStability)	Predicted Stability Change	Effect on Protein Structure	ACMG	ACMG Supporting Criteria
FXI	p.Leu306Pro	Exomes: Not found Genomes: Not found	No data available	−0.32 kcal/mol	Destabilizing	Disallowed phi/psi	VUS	PM2, PP3, PP2
FXI	p.Trp515Gly	Exomes: Not found Genomes: Not found (cov: 31.9)	No data available	−2.94 kcal/mol	Destabilizing	Disrupts all H-bonds Expansion of cavity	VUS	PM2, PP3, PP2

## Data Availability

The data presented in this study are available on request from the corresponding author.

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
