# Peer review of "Clinical and Genetic Heterogeneity of Factor XI Deficiency: Insights from a Southern Italian Cohort"

_ijms, 2025, doi:10.3390/ijms26188807_

Round 1

Reviewer 1 Report

Comments and Suggestions for Authors

Santacroce R et al presented a very nicely written and informative manuscript on bleeding predictability in patients with FXI deficiency. The main conclusions of the paper confirmed results of previous studies emphasizing the clinical and genetic heterogeneity of FXI deficiency, and the lack of strong associations between genotype and FXI:C levels, as well as FXI:C levels and individual bleeding types.

The novelties of the manuscript are that (1) this is the first description of a comprehensive clinical and laboratory analysis of FXI-deficient patients from Southern Italy; and (2) authors found and characterized three novel variant mutations in the F11 gene. It can also be regarded an important new observation that the novel p.Ala561Asp variant segregated in cis position with the p.Glu315Lys variation in one of the index patients and her mother. The associated FXI:C levels in this patient pair and in other patients carrying the p.Glu315Lys variation suggested that p.Ala561Asp in cis position does not apparently influence either the FXI:C level or the clinical presentation.

In the last paragraph of Discussion (from line 311), authors refer to risk stratification models to be used for predicting more accurately the bleeding risk of patients with FXI deficiency. However, they don’t demonstrate the value of the referred risk stratification models. This way the conclusion written in the Abstract, i.e. “comprehensive risk assessment models … are necessary for optimal patient management’ (lines 38-40) is not strongly supported and should be rephrased more cautiously.

Minor remarks:

Gene symbols are usually written in italic characters, i.e. F11 (line 22 and later in the text).

Line 120: No hyphen is necessary in “di-luted” in this position.

Author Response

Comment 1: In the last paragraph of Discussion (from line 311), authors refer to risk stratification models to be used for predicting more accurately the bleeding risk of patients with FXI deficiency. However, they don’t demonstrate the value of the referred risk stratification models. This way the conclusion written in the Abstract, i.e. “comprehensive risk assessment models … are necessary for optimal patient management’ (lines 38-40) is not strongly supported and should be rephrased more cautiously.

Response 1: we thank the Reviewer for her/his helpful comment. We agree that the sentence was wrongly formulated and can be misleading. We erased from the Abstract the sentence and rewritten the last paragraph of Discussion (from line 348).

Minor remarks:

Comment 1: Gene symbols are usually written in italic characters, i.e. F11 (line 22 and later in the text).

Response 1: the FXI gene symbol is now presented in italic characters

Comment 2: Line 120: No hyphen is necessary in “di-luted” in this position.

Response 2: the typo has been corrected.

Reviewer 2 Report

Comments and Suggestions for Authors

This manuscript addresses the clinical and genetic variability of Factor XI deficiency in a Southern Italian cohort. The Authors present clinical symptoms, perform coagulation tests, perform Sanger sequencing to detect genetic variants in F11 gene, as well as in silico predictions of pathogenicity and flexibility. A significant number of well-characterized patients are included for a rare disorder. Three novel F11 gene variants have been identified and bioinformatics tools have been used for variant interpretation. However, the Authors rely mostly on descriptive claims, and assume genotype–phenotype correlations and bleeding predictability without performing any statistical tests or standardized scoring. Besides, there is very little novelty in the conclusion that there is poor genotype–phenotype correlation between FXI and bleeding risk, and perhaps a compensatory coagulation mechanism should be investigated to increase the interest.

Major points

  1. The title is misleading and should be changed. The term “Bleeding predictability” suggests a tool or statistical method for bleeding prediction, while the study shows that FXI:C levels do not predict bleeding. The title should be precise and accurately reflect the study findings. The reviewer suggests: “Clinical and Genetic Heterogeneity of Factor XI Deficiency: Insights from a Southern Italian Cohort”.
  2. The authors conclude that there is no genotype–phenotype correlation, but no statistical testing is presented to support this. Fishers exact or chi-square tests should be performed to assess whether bleeding symptoms are significantly associated with specific variants or genotype (heterozygous vs. compound heterozygous/homozygous). This would strengthen the validity of conclusions.
  3. The assessment of bleeding symptoms is based on descriptive clinical notes rather than a standardized scoring system. Validated quantitative bleeding score should be used to objectively address bleeding severity across the cohort.
  4. Although many family members have been included in this study, the variant segregation with phenotype is not fully explored. Table 1 includes valuable family data on genetic variants, FXI activity, and bleeding symptoms, but the Authors do not discuss whether the variants segregate with the clinical phenotype. The Reviewer recommends highlighting whether bleeding symptoms consistently occur in individuals carrying the same variant and if any asymptomatic carriers are present.
  5. Functional validation of novel F11 variants is missing.
  6. Exploring compensatory mechanisms of the coagulation pathway could explain phenotype variability in FXI patients.

Minor points

  1. In Materials and Methods, the Authors write ”A total of 93 individuals belonging to independent families and additional unrelated 83 index patients with congenital FXI deficiency”, whereas in Results the Authors state “In the present study was investigated a cohort of 93 individuals (37 men and 56 women) with a diagnosis of FXI deficiency from 39 unrelated families” – is the latter after the exclusion criteria? The Authors need to be more precise.
  2. Clearly mark which variants are novel in figure 1 and table 1.

Author Response

Major points

Comment 1: The title is misleading and should be changed. The term “Bleeding predictability” suggests a tool or statistical method for bleeding prediction, while the study shows that FXI:C levels do not predict bleeding. The title should be precise and accurately reflect the study findings. The reviewer suggests: “Clinical and Genetic Heterogeneity of Factor XI Deficiency: Insights from a Southern Italian Cohort”.

Response 1:  we thank the Reviewer for her/his helpful suggestion. The title has been changed as suggested.

Comment 2: The authors conclude that there is no genotype–phenotype correlation, but no statistical testing is presented to support this. Fishers exact or chi-square tests should be performed to assess whether bleeding symptoms are significantly associated with specific variants or genotype (heterozygous vs. compound heterozygous/homozygous). This would strengthen the validity of conclusions.

Response 2: we thank the Reviewer for her/his helpful suggestion that greatly improved the quality of our work. Accordigly, we performed statistic analyses (Fisher exact tests, Student's T test, and correlation analysis) and added a new specific paragraph to strengthen the validity of conclusions (results section, lines 272-293).  

Comment 3: The assessment of bleeding symptoms is based on descriptive clinical notes rather than a standardized scoring system. Validated quantitative bleeding score should be used to objectively address bleeding severity across the cohort.

Response 3: we thank the Reviewer for her/his helpful suggestion that greatly improved the quality of our work. We calculated the BAT score for each FXI defective case (now presented in the last column of the table 1). This allowed for a more objective analysis of bleeding severity (Result section, lines 272-293). 

Comment 4: Although many family members have been included in this study, the variant segregation with phenotype is not fully explored. Table 1 includes valuable family data on genetic variants, FXI activity, and bleeding symptoms, but the Authors do not discuss whether the variants segregate with the clinical phenotype. The Reviewer recommends highlighting whether bleeding symptoms consistently occur in individuals carrying the same variant and if any asymptomatic carriers are present.

Response 4: we thank the Reviewer for her/his helpful suggestion that greatly improved the quality of our work. We analyzed the cohort as suggested and the presence of asymptomatic carriers of F11 gene variant is now presented (Results section, lines 279-286). 

Comment 5: Functional validation of novel F11 variants is missing.

Response 5: We agree with the reviewer that an abnormality in a functional assay significantly enhances the role of the identified variant and allows for better characterization. However, we do not have specific facilities to perform such assays. It is worth noting that many of the previously reported potentially harmful F11 gene variants lack functional validation.

Comment 6: Exploring compensatory mechanisms of the coagulation pathway could explain phenotype variability in FXI patients.

Response 6: we thank the Reviewer for her/his helpful suggestion that greatly improved the quality of our work. The presence of compensatory mechanisms has been discussed and is included in this version of the manuscript (Discussion section, lines 341-347).

Minor points

Comment 1: In Materials and Methods, the Authors write ”A total of 93 individuals belonging to independent families and additional unrelated 83 index patients with congenital FXI deficiency”, whereas in Results the Authors state “In the present study was investigated a cohort of 93 individuals (37 men and 56 women) with a diagnosis of FXI deficiency from 39 unrelated families” – is the latter after the exclusion criteria? The Authors need to be more precise.

Response 1: we agree with the Reviewer that sentences were wrongly formulated and can be misleading. We have rewritten both sentences in the Materials and Methos section (lines 78-80) and in the Results section (lines 149-151). 

Comment 2: Clearly mark which variants are novel in figure 1 and table 1.

Response 2: new F11 gene variants identified in this report have been marked in Table 1 and in Figure 1.

Round 2

Reviewer 2 Report

Comments and Suggestions for Authors

The Authors have answered reviewers' points adequately and included additional tests, statistical analysis and improved the discussion section, which significantly improved the quality of the manuscript.